# Isolation and Functional Analysis of *MbCBF2*, a *Malus baccata* (L.) Borkh CBF Transcription Factor Gene, with Functions in Tolerance to Cold and Salt Stress in Transgenic *Arabidopsis thaliana*

**DOI:** 10.3390/ijms23179827

**Published:** 2022-08-29

**Authors:** Xingguo Li, Xiaoqi Liang, Wenhui Li, Anqi Yao, Wanda Liu, Yu Wang, Guohui Yang, Deguo Han

**Affiliations:** 1Key Laboratory of Biology and Genetic Improvement of Horticultural Crops (Northeast Region), Ministry of Agriculture and Rural Affairs/National-Local Joint Engineering Research Center for Development and Utilization of Small Fruits in Cold Regions/College of Horticulture & Landscape Architecture, Northeast Agricultural University, Harbin 150030, China; 2Horticulture Branch of Heilongjiang Academy of Agricultural Sciences, Harbin 150040, China

**Keywords:** *Malus baccata* (L.) Borkh, *MbCBF2*, cold stress, salt stress

## Abstract

CBF transcription factors (TFs) are key regulators of plant stress tolerance and play an integral role in plant tolerance to adverse growth environments. However, in the current research situation, there are few reports on the response of the CBF gene to Begonia stress. Therefore, this experiment investigated a novel CBF TF gene, named *MbCBF2*, which was isolated from *M. baccata* seedlings. According to the subcellular localization results, the MbCBF2 protein was located in the nucleus. In addition, the expression level of *MbCBF2* was higher in new leaves and roots under low-temperature and high-salt induction. After the introduction of *MbCBF2* into *Arabidopsis thaliana*, the adaptability of transgenic *A**. thaliana* to cold and high-salt environments was significantly enhanced. In addition, the high expression of *MbCBF2* can also change many physiological indicators in transgenic *A**. thaliana*, such as increased chlorophyll and proline content, superoxide dismutase (SOD), peroxidase (POD) and catalase (CAT) activity, and reduced malondialdehyde (MDA) content. Therefore, it can be seen from the above results that *MbCBF2* can positively regulate the response of *A**. thaliana* to low-temperature and osmotic stress. In addition, *MbCBF2* can also regulate the expression of its downstream genes in transgenic lines. It can not only positively regulate the expression of the downstream key genes *AtCOR15a*, *AtERD10*, *AtRD29a/b* and *AtCOR6.6/47*, related to cold stress at low temperatures, but can also positively regulate the expression of the downstream key genes *AtNCED3*, *AtCAT1*, *AtP5CS*, *AtPIF1/4* and *AtSnRK2.4*, related to salt stress. That is, the overexpression of the *MbCBF2* gene further improved the adaptability and tolerance of transgenic plants to low-temperature and high-salt environments.

## 1. Introduction

Plants are constantly affected by various environmental pressures throughout their lives. These environmental factors include biological and abiotic factors, which bring great challenges to the growth and survival of plants [1]. Soil salinization, high-temperature, low-temperature, drought and other abiotic stresses often cause certain oxidative damage to plants, and seriously affect the yield and quality of crops [2,3,4]. Especially under cold conditions, the synthesis and accumulation of various substances in plants are affected, and various life activities and reactions will also undergo dramatic changes. For example, they lead to damage to cell membranes, causing the increased extravasation of soluble substances in cells, and disrupting the balance of substances inside and outside plant cells. The normal life activities of plants are even more seriously threatened. Therefore, the harm of low-temperature stress on plants is in need of attention [5,6]. In order to adapt to these environmental conditions that are not conducive to normal life activities, and to reduce the damage caused by stress, plants form a complex signal transduction system, which can sense and transmit signals, respond to these signals, and activate the expression of relevant genes, such that plants can adapt to the changing environment [7,8,9]. Transcription factors (TFs) play a key role in this process; they can not only sense abiotic stress signals but also specifically activate a number of target genes related to stress response, further enhancing the plant’s tolerance to stress [10,11].

CBF/DREB TFs have AP2 conserved domains, and belong to the plant AP2/ERF TF family. As a key hub of the plant cold stress tolerance mechanism, CBF/DREB TFs can bind to CRT/DRE cis-acting elements in downstream gene promoters to activate downstream cold protein gene expression, resulting in increased proline and chlorophyll content, thereby enhancing the plant’s tolerance to various adversities. In *Arabidopsis thaliana*, the DREB subfamily is further divided into six subgroups, namely the A1–A6 subgroups. Among them, A1 and A2 are the two largest subgroups, including *DREB**1**/CBF* (*CBF1**–**6*) and *DREB2* genes. Genes from each group participate individually or cooperatively in plant responses to different abiotic stresses. For example, A1 subgroup genes are mainly involved in abiotic stresses such as drought and low temperatures [12,13], A2 subgroup genes are mainly induced by high-temperature and water stress [14]; the transcription factors (TFs) of the A1 and A2 subgroups jointly participate in non-Abscisic Acid (ABA)-dependent stress response pathways [15]. The DREB A6 subgroup gene MS DREB6.2 found in apples is involved in regulating the development of adventitious roots and the response to drought stress [16]. In addition, various CBF TFs also play different roles; according to the function of their genes, DREB A1 group members in *A. thaliana* were divided into two groups: CBF1/DREB1C, CBF2/DREB1B, and CBF3/DREB1, which show the fastest up-regulation in response to low-temperature stress [17,18], and the second category is CBF4/DREB1D, DDF1/DREB1F and DDF2/DREB 1E.

More and more studies have shown that CBF/DREB TFs, as key regulators of plant stress tolerance, are widely involved in plant responses to stress [19,20,21]. Previous studies have shown that CBFs can stimulate and induce the expression of the *COR* gene by combining COR elements in the COR gene promoter, thereby making plants show tolerance to cold stress. It was found that a class of ICE1 TFs can specifically bind the MYC element in the promoter of the *CBF3* gene to make the *CBF3* gene express specifically, but they do not show such effects on *CBF1* and *CBF2* [22]. In most plant species, the overexpression of the CBF gene can enhance their cold tolerance, such as for rice, corn, and *B. napus*, et al. [23,24,25]. However, in grapes, the *CBF4* gene is induced by cold stress, while *CBF1*, *CBF2* and *CBF3* are easier to induce by drought [26]. In addition, CBF TF also plays an important role in the salt stress response. *MtCBF4* overexpression in transgenic *A. thaliana* activates the expression of downstream genes, which contain DRE elements, enhancing the tolerance of plants to drought and salt stress [27]. Alonsoblanco et al. [28] found that the cold tolerance effect of *A. thaliana* quantitative trait loci (QTL) was related to the expression of *CBF2* and *CBF* target genes. In addition to *A.*
*thaliana*, many studies have also analyzed the *CBF2* gene of other species such as *Camellia sinensis*, *Jatropha curcas*, and *Solanumly*
*copersicum*, et al. These results all show that CBF2 can improve plant adaptation to high-salt, cold and drought environments [29,30].

The signal transduction pathway requires the interaction of multiple genes. From sensing stress signals to expressing stress response genes, the combination and regulation of various TFs and cis elements are required in order to complete the response of plants to stress. Therefore, the way in which TFs regulate plant stress tolerance genes has become a hot research topic. Previous studies have shown that the loss of function or the mutation of *CBF2* will lead to an increase in the expression of *CBF1* and *CBF3*, indicating that the expressions of *CBF1* and *CBF3* were negatively regulated by *CBF2* [31]. The study found that under cold conditions, the overexpression of *MYB15* reduced the tolerance of plants to cold, while the loss of *MYB15* function increased the expression of *CBF1*, *CBF2* and *CBF3* in plants, and improved the tolerance of plants to freezing stress. These results suggest that *MYB15* plays a role as an upstream regulatory gene of CBF [32]. Xiong et al. [33] found that the transcript levels of *CBF2* and *CBF3* in WT plants were significantly lower than those in *HOS2* mutant plants under cold treatment, suggesting that *HOS2* may act upstream of CBFs. In addition, CBF TFs can further regulate the transcription and expression of downstream genes, and can further enhance the tolerance of plants to low-temperature and salt stress. So far, it has been proven that many downstream genes responding to low-temperature stress and salt stress are regulated by CBF TFs, such as *RD29a*, *COR15a*, *SnRK2.4*, and so on. Studies have shown that *COR15a* and *COR15b* promoters were induced by cold stress in potato and tobacco [34]. A great deal of the downstream functional genes *KIN1*, *COR6.6*, *COR15a*, *RD17*, *RD29a* and *ERD10* located in *A. thaliana* can also be activated by the overexpression of the *CBF/DREB* gene under cold stress [35]. Du et al. [36] found that the expression level of jasmonic acid (JA) biosynthesis genes such as *OsAOS*, *OsOPR1*, *OsAOC*, and *OsLOX2* in rice increased under the induction of a low temperature, which led to the rapid increase of the endogenous JA level. SFR6 protein is a regulator located downstream of CBF. Wathugala et al. [37] found that SFR6 protein was not only related to low-temperature stress in plants but also regulated the expression of defense genes in salicylic acid (SA) and JA-dependent signal transduction pathways. SA and JA are plant hormones that mainly regulate plant development and improve plant stress tolerance [38,39].

Changes in stress-related physiological indicators can intuitively reflect the degree of damage to plants caused by adverse growth conditions. For example, MDA, one of the products of plant plasma membrane peroxidation, can cause serious damage to enzyme proteins and cell membrane systems in plant cells [40]. Therefore, the degree of peroxidation of the plant cell membrane can be reflected by the content of MDA. The higher the MDA content, the higher the degree of oxidation and the greater the damage to the plant. Chlorophyll is one of the crucial factors for the photosynthesis of plants. However, chlorophyll biosynthesis was clearly inhibited under cold stress, and the content of it will be greatly reduced [41,42], which affects the growth and development of plants. In addition, antioxidant enzymes such as SOD, POD, CAT and other enzymes in plants can effectively remove or inhibit peroxide ions in plant cells in a timely manner, thereby reducing the damage caused by stress to plants [43,44,45]. In addition, when the plant is stressed, proline will accumulate in a large amount. Therefore, its content can also be used as an index of plant tolerance to adverse environments [46].

However, there are few CBF TFs studies targeting *Malus* plants. Therefore, the signaling/regulatory pathway of CBF in the stress response of *Malus* plants is still unclear. In this study, the CBF TF gene *MbCBF2* induced by low-temperature and salt stress was found through gene cloning. The main function of this CBF TF gene under low-temperature and high-salt conditions was verified through stable genetic transformation, and the expression of several key downstream genes under relevant stress conditions was identified. The deep-seated causes of cold tolerance and salt tolerance in mountain stator were preliminarily clarified, and were also prepared for the screening of the cold and salt tolerance of new *Malus* materials.

## 2. Results

### 2.1. Cloning and Sequence Analysis of MbCBF2

Prot Param analysis showed that *MbCBF2* encoded 220 amino acids, and its ORF was 663 bp (Appendix A). The theoretical isoelectric point (pI) of the *MbCBF2* protein was predicted to be 5.18, the theoretical molecular mass (MW) was 24.206 kDa, and the average hydrophilicity coefficient was −0.456. Among them, Ala (14.5%), Ser (10.0%), Glu (7.7%), and Pro (7.3%) account for a large proportion. Sequence analysis indicated that the sequence included a DNA binding site and a conserved sequence of AP2 consisting of 58 amino acids, which showed that *MbCBF2* belonged to the AP2 family.

### 2.2. Phylogenetic Relationship of MbCBF2

In order to probe the evolutionary relationship between plant CBF proteins, the CBF proteins of *MbCBF2* and 11 other different species were compared by DNAMAN. It can be seen from Figure 1A that the conserved amino acid sequences of the *MbCBF2* gene are framed in red. The conserved fragments of the CBF TFs of different species had slight differences, but the sequence was roughly the same; in other words, the CBF amino acid sequence of different species had a higher homology in its conserved regions, while the homology in other regions was relatively low.

According to the homologous phylogenetic tree (Figure 1B), MbCBF2 (*Malus baccata*, TQE10660.1) and PdCBF4 (*Populus*
*davidiana*, KF880602.1) belong to the same evolutionary branch, indicating that they have the highest homology and the closest evolutionary relationship, followed by FaCBF4 (*Fragaria ananassa*, AEK94313.1) homogenized in the first cluster of the phylogenetic tree. *MdCBF4* (*Malus domestica*, ART85561.1), MdCBF2 (*Malus domestica*, AGL07698.1), MsCBF3 (*Malus sieversii*, ARO50175.1), and MsCBF4 (*Malus sieversii*, AFU52632.1) were all clustered in the second cluster of the phylogenetic tree, indicating that their kinship is also relatively close. AtCBF3 (*A**. thaliana*, ABV27154.1), AtCBF2 (*A**. thaliana*, ABV27118.1), BnCBF7 (*Brassicanapus*, AAM18959.1), LpCBF3 (*Lepidium perfoliatum*, AGY36892.1), AaCBF3 (*Ageratina adenophora*, AIT39763.1), and MbCBF2 (*Malus baccata*) had a low homology, and were located in the third cluster of the evolutionary tree.

### 2.3. Subcellular Localization of MbCBF2 Protein

The specific location of *MbCBF2* protein was determined by constructing the fusion expression vector of green fluorescent protein (GFP) and the *MbCBF2* gene. It can be seen from Figure 2 that the *MbCBF2*-GFP fusion protein can only be detected in the nucleus (Figure 2E), while the GFP—as the control—can be observed in the whole cytoplasm (Figure 2B). At the same time, the location of the nucleus was confirmed by 4′, DAPI staining (Figure 2F). Therefore, it can be concluded that *MbCBF2* is a nucleus-located protein.

### 2.4. Expression Analysis of MbCBF2 in M. baccata

When the plants were in the control condition (CK), the *MbCBF2* gene had the highest expression in new leaves, followed by roots, while the expression was lower in stems and mature leaves. This indicated that the organs in which the *MbCBF2* gene produces an effect are mainly related to nutrient transport, such as roots and new leaves (Figure 3A).

Under the treatment with high salt, cold, drought and high temperature, the expression of the *MbCBF2* gene in the new leaves of *M. baccata* showed a tendency to increase and then decrease. After treatment with cold, high salt, drought and high temperature for 9 h, 12 h, 6 h and 6 h, respectively, the expression level of the *MBCBF2* gene reached its highest (Figure 3B). The expression trend of *MBCBF2* in the roots was generally consistent with that in new leaves, and reached the highest expression at 12 h, 9 h, 2 h and 6 h, respectively, and then decreased slightly (Figure 3C). The results showed that in the new leaves and roots of the *M. baccata*, these four stresses can cause the up-regulation of *MbCBF2* expression.

### 2.5. Overexpression of MbCBF2 in A. thaliana Enhances Low-Temperature Tolerance

It can be seen from the results of qPCR that *MbCBF2* was very sensitive to low-temperature and high-salt stresses. Therefore, controlled by the CaMV 35S promoter, the *MbCBF2* gene was transformed into *A. thaliana* for verification. Using WT and unload line (UL, *A.*
*thaliana* plants with an ‘empty vector’ after genetic transformation) as controls, the results of RT-PCR in all T_2_ generation transformation lines showed that there were target fragments in 6 transformation lines (S1, S2, S3, S4, S5 and S6) (Figure 4A).

It can be seen from Figure 4B that under the control conditions (low-temperature for 0 h), the phenotypes of all of the *A. thaliana* strains (WT, UL, S1, S4 and S5) were basically the same. However, after 10 h of treatment at −4 °C, the WT, UL, and transgenic (S1, S4 and S5) lines were injured to varying degrees, among which UL and WT had more obvious leaf wilt, and the leaves were seriously damaged. Compared with UL and WT, although some of the leaves of transgenic plants were injured, the overall wilting situation was not particularly obvious.

After 3 days of recovery, the survival rate was calculated. Under control conditions, the survival rates of all *A. thaliana* lines (WT, UL, S1, S4 and S5) were basically the same. However, when treated at −4 °C for 10 h, the survival rates of WT and UL were only 20.53% and 19.62%, while the S1, S4 and S5 transgenic plants survived better; their survival rates were 60.63%, 60.75% and 57.33%, respectively. After growing at low-temperatures for a period of time, more transgenic plants survived than WT and UL (Figure 4C), indicating that the transgenic plants have better low-temperature tolerance.

In the control conditions, the UL, WT and transgenic *A. thaliana* lines showed no significant differences among all of the indicators. After low-temperature treatment, the contents of proline, chlorophyll and MDA, and the activities of SOD, CAT and POD changed, compared with the transgenic lines, the MDA contents of WT and UL were higher, but other indicators were lower than transgenic *A. thaliana* (Figure 5). These datas indicated that transgenic *A. thaliana* plants had stronger antioxidant capacity than UL and WT, and that they can scavenge the intracellular reactive oxygen species (ROS) whilst better preventing plasma membrane peroxidation; thereby, the resilience of transgenic *A. thaliana* to cold stress was improved.

### 2.6. Expression Analysis of Cold-Resistant Downstream Genes in MbCBF2-OE A. thaliana

*A. thaliana* triggers cold stress tolerance through the translation of CBF as an early cold response TF. Therefore, under cold treatment the expression changes of six key genes located downstream of the CBF TFs were analyzed (Figure 6), including *AtRD29a* (*A. thaliana*, D13044.1) [47], *AtCOR6.6* (*A. thaliana*, NM_121602.4) [48], *AtCOR15a* (*A. thaliana*, NM_129815.5) [47], *AtERD10* (*A. thaliana*, BG732218.1) [48], *AtCOR47* (*A. thaliana*, NM_101894.4) [47], and *AtRD29b* (*A. thaliana*, NM_001249003.2) [49]. After the plants were treated at a low temperature of −4 °C for 10 h, the expression levels of six genes in the WT and UL lines were significantly lower than those in the *MbCBF2* transgenic lines, suggesting that the *MbCBF2* can promote the up-regulation of the expression of *AtRD29a*, *AtCOR6.6*, *AtCOR15a*, *AtERD10, AtCOR47* and *AtRD29b*, so as to enhance the tolerance of the plants to low-temperature stress.

### 2.7. Overexpression of MbCBF2 in A. thaliana Improved High-Salt Tolerance

In order to explore the role of *MbCBF2* when the plants are in a high-salt environment, WT, UL and transgenic *A. thaliana* strains (S1, S4 and S5) were transplanted into plastic square pots with a substrate (nutrient soil and vermiculite: 1:1); then, the square pots were placed into a tray, and the plants were irrigated with 200 mM NaCl through the holes at the bottom of the pot. After 7 d, the phenotype of each strain was observed (Figure 7A). Under the salt stress treatment, the leaves of the WT, UL and transgenic plants were all yellowed. Compared with the transgenic plants, the UL and WT leaves’ yellowing was more serious. The survival rate of each *A. thaliana* was counted after 3 days of recovery, as shown in Figure 7B. Under the control conditions, all of the *A.*
*thaliana* strains (WT, UL, S1, S4 and S5) have basically the same survival rate. However, under salt stress, the survival rates of the transgenic plants were 54.87%, 52.78% and 51.99%, respectively, while UL was 17.98% and WT was 18.04%. It is obvious that transgenic *A. thaliana* has a higher survival rate than UL and WT in a high-salt environment.

In addition, the physiological indexes of all of the *A. thaliana* under normal and different concentrations of salt stress were measured. When growing under normal conditions, there were almost no differences in the physiological indexes of any of the *A. thaliana*, but after salt stress treatment, the contents of MDA and proline, and the activities of CAT, POD and SOD of transgenic *A. thaliana* were significantly higher than those of WT and UL, while the content of chlorophyll was significantly lower than that of WT and UL (Figure 8). The results showed that due to the expression of the *MbCBF2* gene, transgenic *A. thaliana* can better adapt to a high-salt environment.

### 2.8. Expression Analysis of Salt Tolerance-Related Genes in MbCBF2-OE A. thaliana

From the analysis results of *MbCBF2* expression, it can be seen that *MbCBF2* can be induced by ABA. The expression of the ABA signal transduction-related gene *AtSNRK2.4*, the ABA synthesis gene *AtNCED3* and gene *PIF1/4* regulating ABA synthesis in *MbCBF2* transgenic *A. thaliana* was further explored. From the analysis results of the expression changes of the downstream genes of six important CBF TFs (Figure 9), *AtCAT1* (*A. thaliana*, NM_118231.4) [50], *AtP5CS* (*A. thaliana*, D32138.1) [51], *AtNCED3* (*A. thaliana*, NM_112304.3) [50], *AtPIF1* (*A. thaliana*, CR457097.1) [52], *AtPIF4* (*A. thaliana*, NM_001337007.1) [52] and *AtSnRK2.4* (*A. thaliana*, AT1G10940) [53], it can be seen that in a normal growth environment, the expression of these genes in all *A. thaliana* strains was almost the same. (Figure 9). However, after 7 days of treatment with 200 mM NaCl, compared with the WT and UL strains, the expressions of *AtCAT1*, *AtP5CS*, *AtNCED3*, *AtPIF1*, *AtPIF4* and *AtSnRK2.4* were significantly higher in *MbCBF2*-overexpression strains. The results made clear that *MbCBF2* could enhance the tolerance of plants to salt stress in two ways.

## 3. Discussion

In recent years, there has been more and more evidence that CBF TF is an indispensable factor in the plant stress response. CBF TFs can regulate the response of plants to drought, low-temperature and high-salt stress. The cold resistance of Forage Maize SAUMZ1 was enhanced by *CBF1* gene [54], while *LeCBF1* [55] helps to improve the ability of *A. thaliana* to adapt to drought/freezing injury. For *MbDREB1* [56], its expression affects the growth state of *A. thaliana* under low temperature, high-salt and drought stress environments, and its overexpression can improve the tolerance of *A. thaliana* to these stresses. In addition, hormones also participate in the regulation of CBFs. Low temperatures can induce JA to remove ICE1/2 inhibition by JAZ1/4, a repressor of the JA signaling pathway, and to positively regulate the transcription of ICE and the expression of *CBF1-3* [57]. The mutation of BIN2, a brassinolide-insensitive factor, enhanced the antifreeze performance of plants [58], while BZR1, BES1 and CES—the downstream TFs of *BIN2*—positively regulated the expression of *A. thaliana* CBFs and the antifreeze performance of plants [58,59].

Sequence homology analysis showed that the ORF of MbCBF2 was 663 bp, and that it encoded 220 amino acids. MbCBF2 protein had a conserved AP2 domain, in which the N-terminal region had nuclear localization signals, and the C-terminal region contains transcriptional activation domains belonging to the CBF family, with the typical structure of the CBF family [60,61,62]. MbCBF2 protein was different from other CBF family proteins in length, but had highly similar conserved sequences, such that it had similar functions, indicating that CBF family genes were highly conserved in the process of evolution. In addition, MbCBF2 protein was hydrophilic, because its average hydrophilic coefficient is −0.456. Through phylogenetic tree analysis, it can be concluded that MbCBF2 protein had the highest homology with PdCBF4 (*Populus davidiana*, KF880602.1) (Figure 1). From the results of subcellular localization, it can be concluded that MbCBF2 is a nuclear localization protein (Figure 2).

Gene expression is specifical in different parts of the plant. Under normal culture conditions, *MbCBF2* was expressed in the new leaves, roots, stems, and mature leaves. Among them, *MbCBF2* had the highest expression in new leaves, followed by roots and stems, and the lowest expression in mature leaves, as shown in Figure 3A. According to the results of real-time quantitative PCR, it can be known that when the new leaves were under the conditions of low temperatures, high salt, drought and high temperatures, the expression level of the *MbCBF2* gene reached its highest at 9 h, 12 h, 6 h and 6 h, respectively (Figure 3B); meanwhile, when the roots were also under such conditions, the expression level of the *MbCBF2* gene reached its highest at 12 h, 9 h, 2 h and 6 h, respectively (Figure 3C). This situation may be due to the fact that new leaves are less sensitive to salt and drought stress than the roots, because the root always suffers from these two stresses first. Compared with the roots, the response of new leaves to low- and high-temperature stress was faster, indicating that new leaves were the first to be stressed in low-temperature and high-salt environments. Therefore, the *MbCBF2* gene can be induced and expressed by cold, high-salt, drought and high-temperature stress. At the same time, it can be seen that the sensitivity of *MbCBF2* to low-temperature and high-salt stress is higher than that for drought and high-temperature stress.

In general, the production and elimination of free radicals maintain a dynamic balance, such that plant cells will not be damaged [63]. However, when plants are under abiotic stress, they will accumulate a large amount of reactive oxygen species in cells. In the short-term period of stress, plants can rely on their own protective enzymes to remove reactive oxygen species. However, over a certain period of time, the amount of reactive oxygen species exceeds the ability of plants to remove reactive oxygen species, which will damage plants [64,65]. The enzyme removal system in plants mainly includes SOD, POD and CAT. When plants are under environmental stress, H_2_O_2_ is converted into reactive oxygen species in order to avoid damage [66]. The amount of MDA can determine the damage degree of the plant cell membrane, and its content is positively correlated with membrane permeability [67]. Compared with Chinese kale plants grown under normal conditions, kale plants under salt stress showed higher levels of MDA [68]. Overexpressed *PeCBF4a* enhanced the antioxidant capacity of transgenic plants and reduced the MDA content, but raised the activities of SOD, CAT and POD so as to reduce the harm to plants brought about by various adverse growth conditions [69]. This experiment studied the effects of *MbCBF2* on plant growth status and growth under low-temperature and high-salt stress. Figure 4B and Figure 7A showed the results of low-temperature and high-salt stress treatment on WT, UL and transgenic plants. UL, that is, *A**. thaliana* without the vector inserted with the *MbCBF2*, was used as a negative control group to evaluate the function and effect of the foreign gene. From the morphological results, compared with the WT, the transgenic plants had stronger adaptability in low-temperature and high-salt environments. The survival rate of *A. thaliana* can be seen in Figure 4C and Figure 7B. The transgenic plants had a higher survival rate, while the survival rates of UL and WT were basically the same, and both were lower than those of the transgenic plants. This result showed that transgenic plants have stronger cold tolerance and salt tolerance than WT and UL plants.

Under low-temperature and high-salt stress, the measured physiological index changes showed similar change trends, as shown in Figure 5 and Figure 8. Compared with the WT and UL, CAT, SOD and POD activities, and chlorophyll, proline and MDA contents, the transgenic plants had significant changes. In addition to the decrease of MDA content, other indicators increased significantly, indicating that transgenic plants had strong vitality under low-temperature and high-salt stress. In addition, the enzyme activity in transgenic plants also increased significantly, which improved the ability of transgenic plants to remove peroxides, thereby reducing the peroxidation of the cell plasma membrane and improving the survival rate of the plants under low-temperature and high-salt conditions. Although the enzyme activity of UL and WT also increased, it was still lower than that of the transgenic plants. At the same time, under low-temperature and high-salt stress, the stability of the chlorophyll content and the increase of the proline content provide a guarantee for the normal life activities of plants.

The expression of stress tolerance genes is affected by a variety of TFs, which play an extremely important role in the response of plants to stress (including biological and abiotic stress). CBF binds to the CRT/DRE cis-acting elements of downstream genes to stimulate the up-regulation of the expression of these genes, thereby enhancing the stress tolerance of the plants. The expression results of cold-responsive genes (*At**COR15a*, *AtRD29a*, *AtCOR6.6*, *AtERD10*, *AtCOR47* and *AtRD29b*) downstream of CBF were analyzed under control and low-temperature treatment conditions. As shown in Figure 6, the results showed that *MbCBF2* TF promoted the expression of *At**COR15a*, *AtRD29a*, *AtCOR6.6*, *AtERD10*, *AtCOR47* and *AtRD29b* more obviously under low-temperature conditions. ABA is the main substance for plants to deal with salt stress. ABA can alleviate the ion stress and osmotic stress caused by excessive salt. Therefore, the water balance and the integrity of the cell membrane structure are maintained. Under high-salt conditions, the expression of the ABA synthesis-related gene *At**NCED3* and the ABA signal transduction-related gene *At**SnRK2.4* were significantly up regulated in transgenic *A.*
*thaliana*. According to this phenomenon, it can be speculated that *MbCBF2* may participate in the response of plants to salt stress by regulating ABA synthesis and signal transduction. As shown in Figure 9, due to the induction of *MbCBF2* expression in transgenic *A. thaliana*, the other salt stress-responsive genes *At**CAT1*, *AtP5CS*, *AtPIF1* and *AtPIF4* were also significantly up-regulated, thereby enhancing the plant adaptation to salt stress.

To sum up, combined with previous studies and the above results, a potential model was derived to describe the mode of action of *MbCBF2* under salt and cold stress (Figure 10). First, salt stress induced the expression of *MbCBF2*, thereby evidently in creasing the expression level of *AtNCED3*, promoting ABA biosynthesis and signal transduction, causing a significant up-regulation of *AtSnRK2.4* expression and enhancing the adaptability of transgenic plants to salt stress. In addition, the expression levels of the key genes *AtCAT1*, *AtP5CS*, *AtPIF1* and *AtPIF4* in response to salt stress were also significantly increased when plants were stressed, indicating that key genes under salt stress could be regulated by *MbCBF2*, thereby enhancing the salt tolerance of transgenic plants. When a cold shock signal is felt, *MbCBF2* is induced by cold stress, and its encoded product specifically binds to the CRT/DRE cis-elements on the promoters of multiple anti-stress genes, causing the high expression of downstream cold-regulated genes (*AtCOR15a*, *AtRD29a*, *AtCOR6*, *AtERD10*, *AtCOR47* and *AtRD29b*), thereby enhancing the cold tolerance of plants. 

## 4. Materials and Methods

### 4.1. Plant Material and Growth Conditions

The tissue culture seedlings of *M. baccata* (*Malus baccata* (L.) Borkh) came from the National Apple Germplasm Resource Garden, and were a cold-resistant genotype with ID DGB0367 grown in propagation medium containing Murashige and Skoog (MS), Agar, and 1 mg/L 6-benzylaminopurine (6-BAP) + 0.5 mg/L indole-3-butyric acid (IBA). After growing for 30 days, the robust tissue culture seedlings were transferred to the rooting medium. The medium used was MS solid medium of 1.2 mg/L IBA. The tissue culture seedlings were grown in the medium for 45 days. The seedlings with thick roots were moved into Hogland nutrient solution to continue to grow, and the nutrient solution was changed every 3 days. Clean and sterilized glass bottles with a height of 20 cm were used for the hydroponics. At the same time, a floating plate was placed at the mouth of the bottle to carry the seedlings for colonization. In total, 2/3 of them were immersed in the nutrient solution, and the environment was kept ventilated during the root seedling raising process. The incubator always maintained a temperature range of 24–26 °C and a relative humidity of about 80% [66]. The stress treatment was carried out according to the method of Han et al. [70]. When 8 to 9 mature leaves were grown (the leaves were fully expanded), the hydroponic seedlings were placed under the conditions of high salt (the small plants were irrigated with Hogland nutrient solution containing 200 mM NaCl), low temperatures (the incubator temperature was set to −4 °C), dehydration (the small plants were irrigated with Hogland nutrient solution containing 20% PEG6000 to simulate drought conditions) and high temperatures (the incubator temperature was 37 °C). The control group consisted of hydroponic seedlings cultured in normal Hoagland nutrient solution. After 0, 2, 6, 9, 12 and 16 h of treatment, root, stem, new leaf and mature leaf samples (100 mg) were randomly collected and sealed, including control and treated plant samples (3 plants were selected, respectively), and were immediately put into liquid nitrogen. Then, for the later extraction of RNA, they were stored at −80 °C [71]. All of the results were obtained from 3 independent experiments on each of the plants at each time point.

### 4.2. Cloning and Quantitative Expression Analysis of MbCBF2

The EasyPure Plant RNA Kit (TransGen Biotech, Beijing, China) was used to extract the total RNA from the new leaves, mature leaves, roots and stems, and the DNA was purified by RNase-Free DNase I purification. We used the HiFiScript gDNA Removal cDNA Synthesis Kit (Kangweishiji, Beijing, China) and Oligo (dT) 18 as the primer for reverse transcription to complete the synthesis of the cDNA First Strands. A pair of specific primers (*MbCBF2*-F and *MbCBF2*-R, Appendix A) were designed and synthesized according to the homologous region of *M**bCBF2*, such that the full-length sequence of cDNA could be amplified. The PCR amplification used cDNA as a template to obtain the target fragment; it was ligated into the pEASY-T1 Cloning Kit (TransGen Biotech, Beijing, China), and positive colonies were transformed, screened and sent for sequencing. According to the viewpoints of Jiang and Zhou, the *MbCBF2* method was analyzed by real-time PCR [72].

The real-time PCR primers q-PCR-F and q-PCR-R were designed based on the conservative part of the *MbCBF2* sequence. The control group was the *Actin* gene, which can be stably expressed under different conditions. *Actin* primers were designed according to the sequences in the GenBank database (*Actin*-F and *Actin*-R). Appendix A showed the primer sequences. The expression level of the *Mb**CBF2* gene was detected using TransStart^®^ Green qPCR Super Mix (TransGen Biotech, Beijing, China) according to the manufacturer’s protocol. The following was the PCR reaction system: 2xMix 12.5 μL, ddH_2_O 9 μL, cDNA 1.5 μL, primer F 1 μL, and primer R 1 μL. The PCR reaction procedure was as follows: 95 °C for 5 min, 95 °C for 45 s, 56 °C for 1 min, 72 °C for 1 min, 35 cycles, and 72 °C for 5 min. We used the 2^−ΔΔCT^ method to analyze the relative transcript level [73,74].

### 4.3. Bioinformatics Analysis of the MbCBF2 Gene

The sequence was analyzed by Blast of the NCBI database (http://www.ncbi.nlm.nih.gov./) (accessed on 1 May 2021), and the CBF protein sequences of other species were downloaded. A series of similar sequences was obtained by Blast in the NCBI database with the amino acid sequence of AtCBF2 (*A**. thaliana*, ABV27118.1), AtCBF3 (*A**. thaliana*, ABV27154.1), AaCBF3 (*Ageratina*
*adenophora*, AIT39763.1), BnCBF7 (*Brassica napus*, AAM18959.1), MdCBF4 (*Malus*
*domestica*, ART85561.1), MdCBF2 (*Malus*
*domestica*, AGL07698.1), *MsCBF3* (*Malus*
*sieversii*, ARO50175.1), MsCBF4 (*Malus*
*sieversii*, AFU52632.1), LpCBF3 (*Lepidium*
*perfoliatum*, AGY36892.1), FaCBF4 (*Fragaria*
*ananassa*, AEK94313.1), and PdCBF4 (*Prunus*
*dulcis*, BBN69788.1), which was selected for the experiment. We used MEGA 7 to perform the multi-sequence alignment analysis and build an evolutionary tree.

### 4.4. Subcellular Localization Analysis of the MbCBF2 Protein

*Sal*I and *BamH*I are two sites of the pSAT6-GFP-N1 vector. *MBCBF2* ORF was cloned between the two sites. The transformation of the *MBCBF2* GFP construct containing modified red-shift GFP at the *Sal*I-*BamH*I site into onion epidermal cells was achieved using particle bombardment technology [75,76]. The nuclear marker in the nuclear detection was DAPI staining. Confocal microscopy observed the transient expression of *MbCBF2*-GFP fusion protein (LSM 510 Meta, Zeiss, Germany).

### 4.5. Construction of Transgenic A. thaliana

Firstly, the restriction sites of *Sal*I and *BamH*I were added at the 5′ and 3′ ends of *MbCBF2* cDNA by PCR to build an expression vector of transformed columbia-0 ecotype *A. thaliana*. In order to construct the pCAMBIA2300-*MbCBF2* overexpression vector, a homologous recombinase (Novizan C115-01/02) was used to join the PCR product with the restriction site to the pCAMBIA2300 vector. The *Agrobacterium*-mediated method was used in the GV3101 transformation, and the *Mb**CBF2* gene was introduced into columbia-0 ecotype *A**. thaliana* by the inflorescence infection method; the transformation of the empty plasmids pCAMBIA2300-*MbCBF2* and pCAMBIA2300 into WT *A. thaliana* was realized [77]. Plants successfully transfected with the *Mb**CBF2* gene on 1/2 MS medium containing 50 mg/L Kanamicin were selected. Through further screening, T_3_ generation plants were obtained and further analyzed.

### 4.6. Determination of the Related Physiological Indicators

The selected WT, UL and transgenic lines (S1, S4, S5) were sown in the medium, respectively, until the *A. thaliana* seedlings grew four leaves; they were then moved to a nutrition bowl and marked. The UL, WT and transgenic plants were divided into three groups: one group was cultured normally at 23 °C, while the other two groups were placed under −4 °C for cold treatment, and were treated with 200 mM NaCl, respectively. After continuous watering for 7 days, we observed the phenotype of *A. thaliana*. Each *A. thaliana* leaf was sampled, and the content of chlorophyll, proline and MDA, and the activities of SOD, POD and CAT were measured [78,79,80,81]. The treated *A. thaliana* were placed in a lighted incubator for normal watering in order to recover; we then observed the phenotype of each *A. thaliana*, and ascertained its survival rate.

### 4.7. Analysis of the Downstream Gene Expression of MbCBF2

We extracted the mRNA of WT, UL and *MbCBF2* transgenic *A. thaliana* grown under normal conditions, low-temperature stress, and salt stress, respectively, and reverse transcribed them into the first-strand cDNA, which was used as a template. With *AtActin* as the internal reference, qPCR tests were conducted on several significant regulatory genes downstream of CBF TF: key genes in response to low-temperature stress (*AtCOR15a*, *AtRD29a*, *AtCOR6.6*, *AtERD10*, *AtCOR47* and *AtRD29*) and salt-related genes (*AtNCED3, AtCAT1, AtP5CS, AtPIF1, AtPIF4* and *AtSnRK2.4*). The specific primers used are shown in Appendix A. The reaction system is the same as that in Section 2.2.

### 4.8. Statistical Analysis

One-way analysis of variance was performed using SPASS data processing system software. We repeated all of the experiments 3 times, and measured the standard error (±SE), respectively. The statistical difference is called significant at * *p* ≤ 0.05 and ** *p* ≤ 0.01.

## 5. Conclusions

In this experiment, a new CBF-TF named MbCBF2 was isolated from *Malus* plants. The MbCBF2 protein had a conserved AP2 domain and a typical structure for the CBF family, as it belonged to the CBF family. According to the subcellular localization results, the MbCBF2 protein was located in the nucleus. According to the phylogenetic tree results, the MbCBF2 protein had the highest homology with PdCBF4. Low-temperature, high-salt, high-temperature and drought stress can all induce the up-regulated expression of the *MbCBF2* gene in *M. baccata*. In addition, under the stimulation of low-temperature and high-salt conditions, the expression of *MbCBF2* in new leaves and roots was higher. In *A.*
*thaliana*, *MbCBF2* responds positively to low-temperature and osmotic stress. After the introduction of *MbCBF2* into *A. thaliana*, the adaptability of transgenic *A. thaliana* to cold and high-salt environments was significantly enhanced. In addition, the high expression of *MbCBF2* also significantly changed many physiological indexes of transgenic *A.*
*thaliana*. Except for the content of MDA, the contents of proline and chlorophyll, and the activities of CAT, POD, and SOD of the WT and UL lines were significantly lower than those of transgenic *A. thaliana*. *MbCBF2* can also regulate the expression of its downstream genes in transgenic lines. After CBF2 binds with cis-acting elements at low temperature, it positively regulates the expression of the key genes *AtCOR15a*, *AtRD29A/B* and *AtCOR6.6/47*, and positively regulates the expression of *AtCAT1*, *AtP5CS* and *AtPIF1/4* under salt stress. It was also found that the expression of the ABA synthesis-related gene *AtNCED3* and the ABA signal transduction-related gene *AtSNRK2.4* were also up-regulated in transgenic *A. thaliana* under high-salt conditions. Therefore, the overexpression of the *MbCBF2* gene further improved the adaptability and tolerance of transgenic plants to low-temperature and high-salt environments. Therefore, the overexpression of the *MbCBF2* gene further improved the adaptability and tolerance of transgenic plants to low-temperature and high-salt environments.

## Figures and Tables

**Figure 1 ijms-23-09827-f001:**
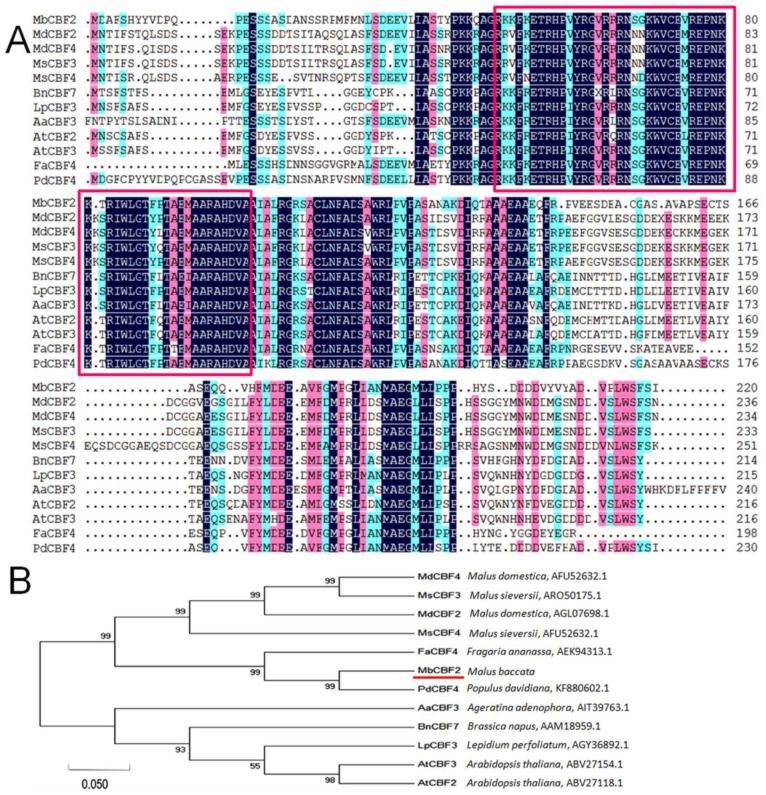
Comparison and phylogenetic relationship between the amino acid sequences of MbCBF2 and the CBF of other species. (**A**) Through comparison, MbCBF2 was compared with CBF transcription factor proteins of other plants. The red box is the conserved sequence of the gene, and the red underline is the target protein. (**B**) Analysis of MbCBF2 and CBF transcription factor proteins in different plants by phylogenetic tree. The MEGA-7 adjacency method was used to construct this tree. AtCBF2 (*A*. *thaliana*, ABV27118.1), AtCBF3 (*A*. *thaliana*, ABV27154.1), AaCBF3 (*Ageratina adenophora*, AIT39763.1), BnCBF7 (*Brassica napus*, AAM18959.1), MdCBF4 (*Malus domestica*, ART85561.1), MdCBF2 (*Malus domestica*, AGL07698.1), MsCBF3 (*Malus sieversii*, ARO50175.1), MsCBF4 (*Malus sieversii*, AFU52632.1), LpCBF3 (*Lepidium perfoliatum*, AGY36892.1), FaCBF4 (*Fragaria ananassa*, AEK94313.1), and PdCBF4 (*Prunus dulcis*, BBN69788.1).

**Figure 2 ijms-23-09827-f002:**
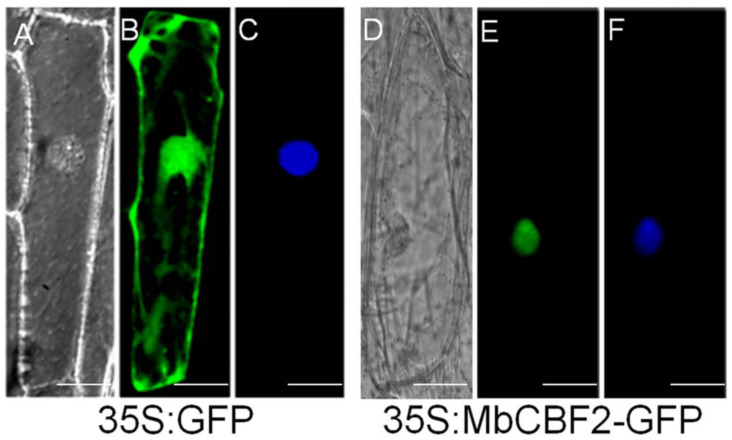
The translation products of 35s-GFP and 35s-*MbCBF2*-GFP were expressed in onion epidermal cells. They were observed under bright light (**A**,**D**) using a fluorescence microscope, a GFP signal image (**B**,**E**) and a DAPI staining image (**C**,**F**) under dark conditions. The scale bar represents 5 μm.

**Figure 3 ijms-23-09827-f003:**
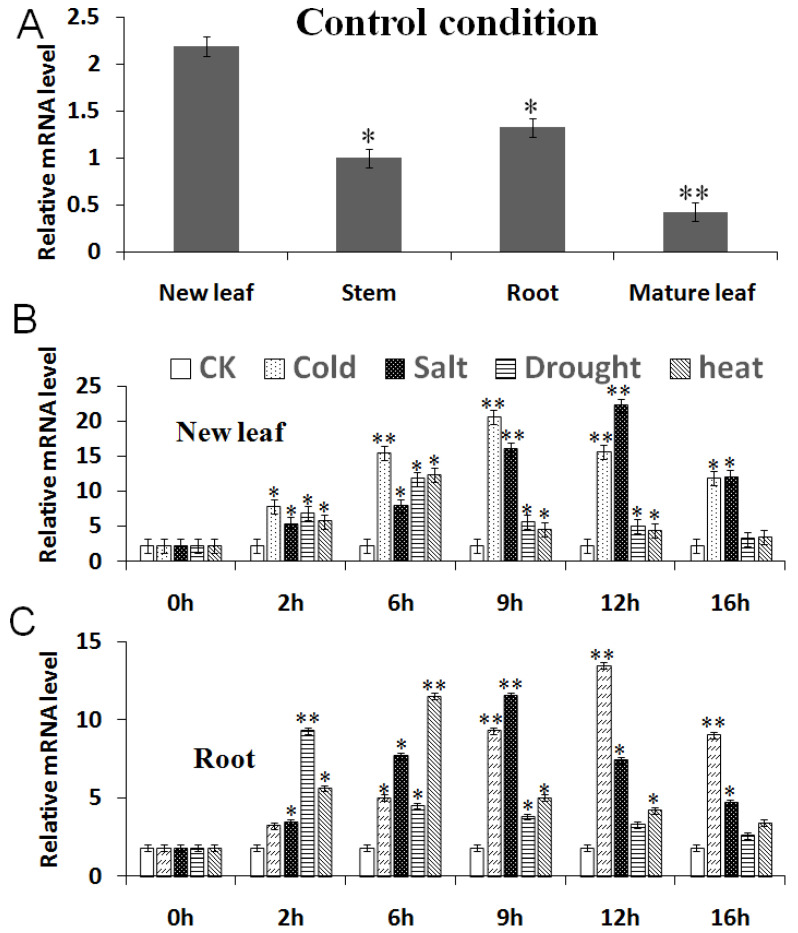
Q-PCR results of the *MbCBF2* gene. (**A**) Expression of *MbCBF2* in new leaves, stems, roots and mature leaves of the normal-growing *Malus*
*baccata*. Expression of the *MbCBF2* gene in (**B**) new leaves and (**C**) roots under the control (CK) and four stress conditions. Each datum in the figure is taken as the average value obtained by repeating three times. Compared with the control group, the asterisks above the column indicate significant differences and extremely significant differences (*, *p* ≤ 0.05; **, *p* ≤ 0.01).

**Figure 4 ijms-23-09827-f004:**
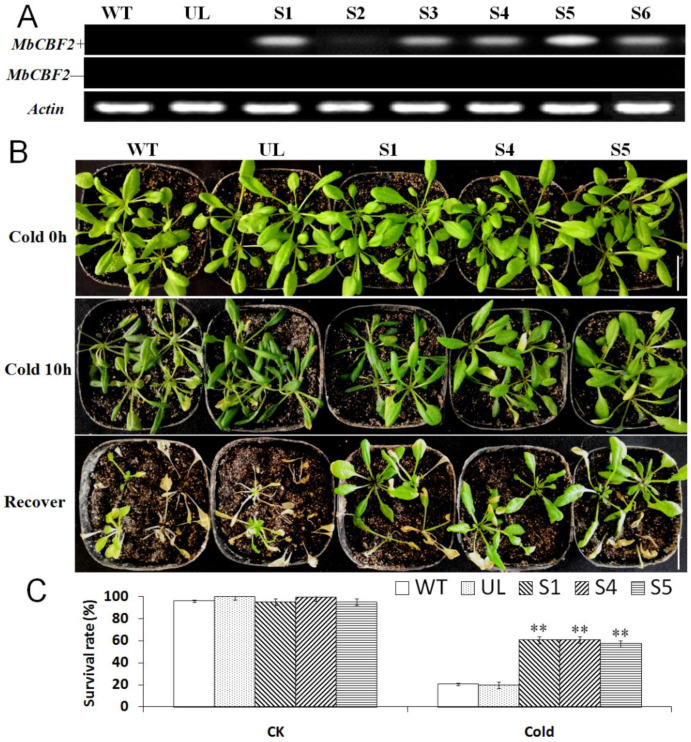
The cold tolerance of *A. thaliana* was improved due to the overexpression of *MbCBF2*. (**A**) The expression levels of *MbCBF2* in WT, UL and T_2_ transgenic of transgenic *A. thaliana* were observed by semi-quantitative RT-PCR with *MbCBF2*-specific primer (*MbCBF2*+) and non-specific primer (*MbCBF2*−). The control was actin. (**B**) Phenotypes of the *MbCBF2* transgenic *A. thaliana* lines (S1, S4, S5), WT and UL under low-temperature stress and recovery. (**C**) A *t*-test showed that there were significant differences in the survival rates between transgenic *A. thaliana* (S1, S4 and S5) and WT lines under low-temperature conditions (**, *p* ≤ 0.01). The scale bar corresponds to 3 cm.

**Figure 5 ijms-23-09827-f005:**
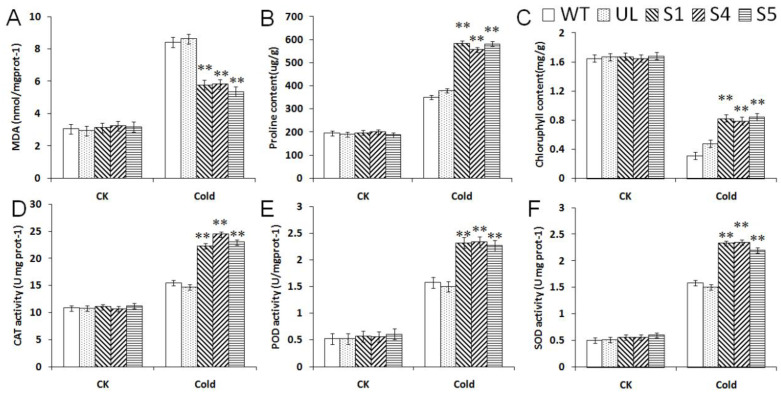
Effect of *MbCBF2* overexpression on the physiological indexes of *A. thaliana* under cold tolerance. (**A**) MDA content, (**B**) proline content, (**C**) chlorophyll content, (**D**) CAT activity, (**E**) POD activity, and (**F**) SOD activity. The means and standard errors of the three repetitions are represented by the data. There was a significant difference between transgenic and WT *A. thaliana* in the treatment group, which was indicated by asterisks above the error bars (**, *p* ≤ 0.01).

**Figure 6 ijms-23-09827-f006:**
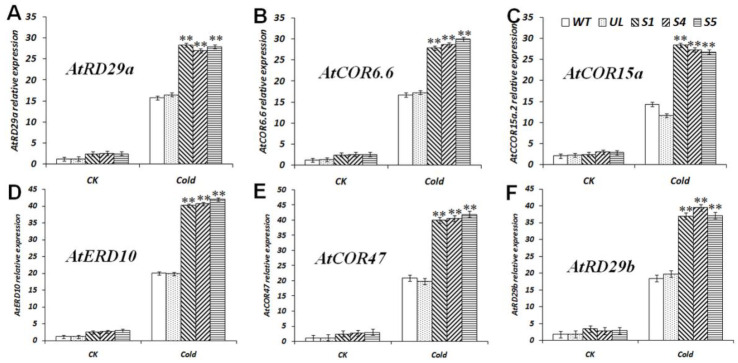
Expression levels of six cold stress-related genes in WT, UL, and transgenic *A. thaliana* under low temperatures. (**A**) Relative expression level of *AtRD29a*, (**B**) relative expression level of *AtCOR6.6*, (**C**) relative expression level of *AtCOR15a*, (**D**) relative expression level of *AtERD10*, (**E**) relative expression level of *AtCOR47*, (**F**) relative expression level of *AtRD29b*. The data are the average of three repetitions. Asterisks above the columns indicate significant differences compared to the WT under cold stress (**, *p* ≤ 0.01).

**Figure 7 ijms-23-09827-f007:**
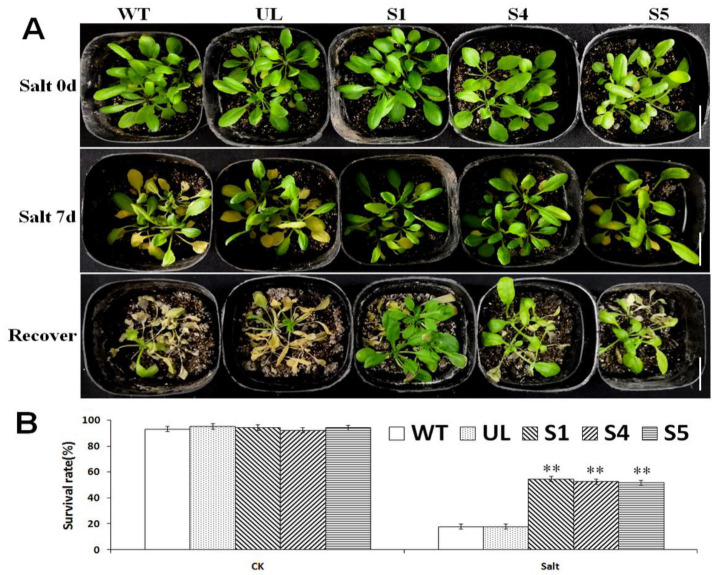
Due to the overexpression of *MbCBF2*, transgenic *A. thaliana* plants can better adapt to a high-salt environment. (**A**) Phenotype of *A. thaliana* under control conditions (Salt 0 d), salt stress (Salt 7 d), and recovery. All of the test lines, including the T_3_ transgenic *A.*
*thaliana* (S1, S4, and S5), WT and UL dealt with salt stress for 7 days, and then with control water management for 3 days for recovery. (**B**) Survival rates of WT, UL and transgenic lines under control and high-salt conditions (200 mM NaCl). The white scale in the figure indicates the actual length of 3 cm. The *t*-test showed that there were significant differences in the survival rates between transgenic *A. thaliana* and the WT lines under salt stress (**, *p* ≤ 0.01).

**Figure 8 ijms-23-09827-f008:**
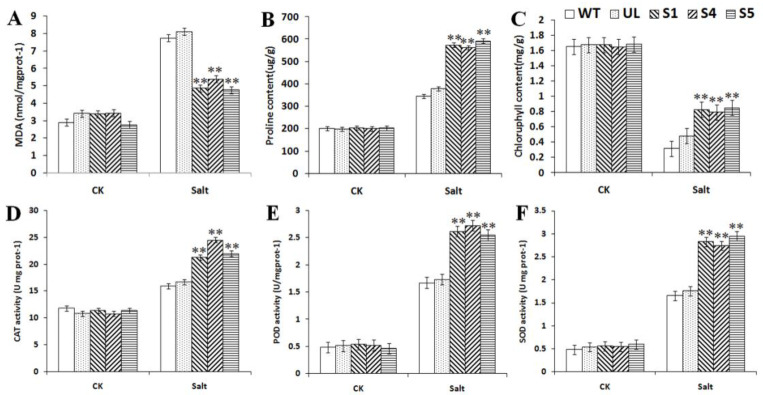
Effect of *MbCBF2* over expression on the physiological indices of *A. thaliana* in a high-salt environment. (**A**) MDA content, (**B**) proline content, (**C**) chlorophyll content, (**D**) CAT activity, (**E**) POD activity, and (**F**) SOD activity. The means and standard errors of the three repetitions are represented by the data. There was a significant difference between transgenic and WT *A. thaliana* in the treatment group, which is indicated by asterisks above the error bars (**, *p* ≤ 0.01).

**Figure 9 ijms-23-09827-f009:**
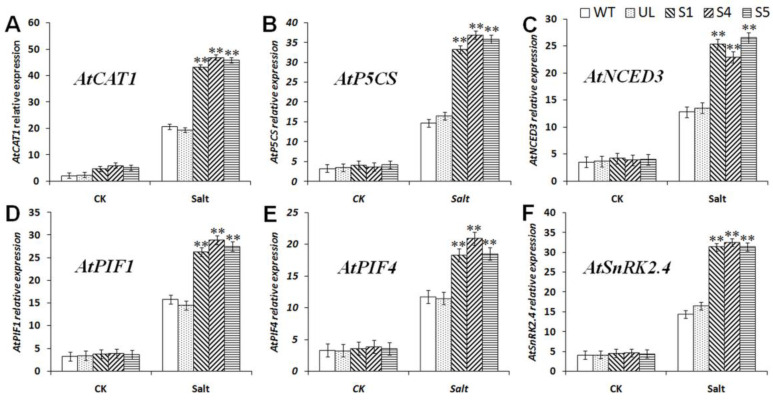
Expression levels of drought stress-related genes in WT, UL, and transgenic *A. thaliana.* under salt stress. (**A**) Relative expression level of *AtCAT1*, (**B**) relative expression level of *AtP5CS*, (**C**) relative expression level of *AtNCED3*, (**D**) relative expression level of *AtPIF1*, (**E**) relative expression level of *AtPIF4*, and (**F**) relative expression level of *AtSnRK2.4*. The data are the average of three repetitions. Asterisks above the columns indicate that there is a significant difference between transgenic and WT *A. Thaliana* under salt stress (**, *p* ≤ 0.01).

**Figure 10 ijms-23-09827-f010:**
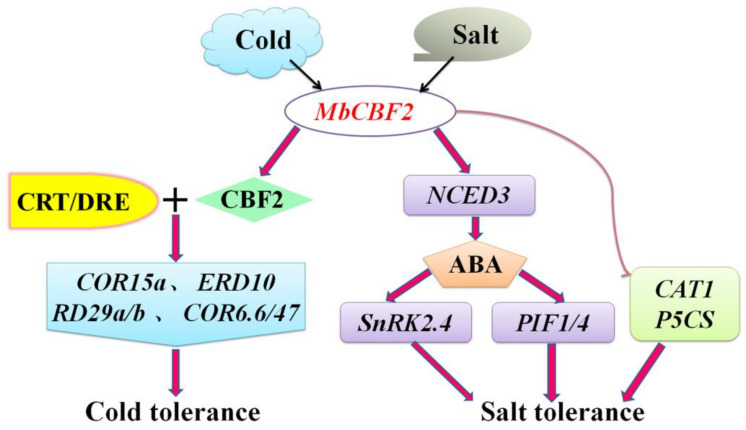
A possible model of *MbCBF2* response to low temperature and salt stress. First, cold stress induces the expression of *MbCBF2*. CBF2 protein specifically binds to the CRT/DRE cis-elements of the cold responsive gene promoter, thereby activating the high expression of downstream genes (*AtCOR15a*, *AtRD29a*, *AtCOR6*, *AtERD10*, *AtCOR47* and *AtRD29b*) and enhancing the cold tolerance of plants. Salt stress induces the expression of *MbCBF2*, thus significantly increasing the expression level of *AtNCED3*, promoting ABA biosynthesis and signal transduction, resulting in the significant up regulation of *AtSNRK2.4* and AtPIF1/4 expression. At the same time, the expression levels of other key genes in response to salt stress, *AtCAT1* and *AtP5CS*, also significantly increased, positively regulating the response of plants to salt stress.

## Data Availability

The original data for this present study are available from the corresponding authors.

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
