# Peer review of "Isolation and Functional Analysis of MbCBF2, a Malus baccata (L.) Borkh CBF Transcription Factor Gene, with Functions in Tolerance to Cold and Salt Stress in Transgenic Arabidopsis thaliana"

_ijms, 2022, doi:10.3390/ijms23179827_

Round 1

Reviewer 1 Report

The submitted paper focused on functional analysis of DREB1/CBF tanscription factor from Malus baccata by developing over-expressor transgenic lines of Arabidopsis thaliana. DREB1/CBF TF specifically interact with cis-acting elements or DRE/CRT containing target genes and control the expression of stress-related genes or stress regulatory genes under normal conditions. The authors determined the expression of various DRE/CRT containing target genes along with physiological indicators of stress tolerance. I have following observations
1. The Title is too lengthy and do not reflect the essence of study.  It neds to be revised. 

2. The language of MS is of moderate or poor and needs necessary rectifications. For example, Line 16-18; 21-22; 24-25; 27; 29; 32; 43-44; 45-46; 49-53; 65; 82-83; 86; 90-91; 92-94; 101-103; 120-122; 123-124; 127; 128-129 .... and so on
3. Introduction: It needs revision as content organization is also poor. Being a reader unable to understand what is the objective of the study. At Line 55, molecuar mechanism of stress sensing and signaling is required. Then describe different components in relation to DREB/CBF. Then highlight upstream or downstream genes (either they directly regulate physiological process) or indirectly mediate physiological process. then Discuss what is unknown. What is the hypothesis of the study. What are the objectives of the study. 

4. Discussion: This part needs to be re-written keeping in view objetives of the study. In most of the studies, over-expression of these genes mediate the expression of other genes (downstream/upstream) or stress regulatory genes even under under normal conditions as higher expression/higher protein of DREB/CBF can recognize DRE/CRT containing target sequences in promoters of target genes under normal conditions. This situation cause metabolic burden on transgenic plants. Various scientists used stress inducible promoters for optimization of expression of this gene. Though I can see greater expression of MbCBF2 genes in transgenic lines under normal conditions Fig 4A, gene expression of other genes tested in this study was normal Fig 6; 9 under normal conditions. In addition, amount proline is also low. However, I am unable to see the construct to assess what type of strategy was used. 
5. Conclusion: Conclusion does not mean key results, but it means key findings. This needs to be comprehensive and in 3-4 lines. 
6. Mechanism Model. NCED3 is ABA biosynthesis related gene and it is working upstream to ABA, while SnRK is working downstream to ABA. This needs to be rectified in Figure. In addition, Model needs what happened under Normal conditions. 

Overall, the MS can be accepted after major revisions suggestd above.  

Reviewer 2 Report

The manuscript IJMS-1878088 deals with very interesting and important topic of CBF2 gene analysis in crab apple, ornamental tree species, Malus baccata. The provided results are very valuable and they were received in wide range of experiments, which can make very important contribution for readers in the area of plant molecular science. However, the presentation of the used methods and received results as well as the writing of this manuscript is poor and needs significant improvement. Therefore, I strongly recommend authors to make major revision of the text and in some parts, to re-write fragments of sub-sections almost completely. If authors agree with this, they must follow major and minor comments and suggestions listed below, and the revised manuscript has to be reconsidering one more time for possible acceptance for the publishing.

Major comments/suggestions:

(1) English has to be improved in entire manuscript. The general requirement includes simple, clear and understandable ‘scientific English’. Below, only a few random examples of non-suitable English are present but there are much more in the whole manuscript. L540: “It can be said…” (Very colloquial. It is better to use ‘It can be concluded…’); L502: “…50 mg/L Kana…” (Is this kanamicin?); L501: “…Select positive plants…” and L466-467 “…positive colonies…” (Plants or colonies cannot be ‘positive’ or ‘negative’. Authors want to say about plants (or colonies) with confirmed insertion of the transgene); L456: “…a refrigerator at -80ºC…” (The name of this equipment is ‘deep-freezer’ or ‘freezer’ for -80ºC but not ‘refrigerator’); L285 (Legend of Figure 7): The first sentence in the legend is absolutely not acceptable for this purpose “As a result of the overexpression of…”; L88 and L229:“…lower tolerance to low temperature…” and “…has higher low temperature tolerance…” (The sense is confusing); L86: “…A. thaliana train…” (Unclear meaning); L71-75: This sentence contains three time word ‘different’ in the first two lines; L50-51. It is better to avoid colloquial terms “to the body” and “in the body” in relation to plants.

(2) The section 4 ‘Materials and Methods’ has to be almost re-written because so much information was missed, not present or sonfusing. I cannot describe all of them because it require a lot of my time but I will just briefly indicating and authors have to re-write the entire section. Sub-section 4.1: “…seedlings of M. baccata…” What is the genotype, origin, accession ID and where it was received from? It has to be available for readers if somebody wishes to reproduce this research. Seedlings were not ‘inoculated’ but grown in medium. MS medium was liquid or with agar? Size of containers? (it would be good to present a photo in Supplementary material but this is not compulsory). L443: The first reagent was missed in the phrase “+ 0.5 mg/L +…”. All abbreviated reagents must be in full name in the first instance. ‘6-BA’ is more commonly known as ‘6-BAP’ and full name is important (‘6-benzylaminopurine’); ‘IBA’ is ‘indole-3-butyric acid’ or not? Many sentences are written as a protocol, which is unacceptable for such manuscript, please modify. Examples: L445: “Select the seedlings with…”; L501: “Select positive plants…”; L517: “Extractedthe mRNA…”; L527: “Repeat all experiments 3 times…”. L448-449: The sentence about salt stress is unclear. Please provide more details about hydroponics, type and size of containers, bubbling or not, how plants were supported, why so strong salt stress was used (this is about 35% of salinity in sea water). Low (high) temperature – did authors use cold (hot) nutrient solution? Circulated or not? Maybe, container was in cold (hot) room? Drought – this is not a ‘drought’ at all because adding of 20% PEG-6000 (full name, please) in hydroponics is a ‘dehydration, which can simulate drought’. L453-455. How many biological replicates (plants) were used in each time-point for sampling and in further results presented in Figures? In L527 it is written about ‘repeated experiments for three times’. Does this mean three plants for each time-point of sampling? Please clarify it either in each sub-section or only in sub-section 4.8 but for all experiments. Fragment in L469-475: This fragment about qPCR is unacceptable in such form. Please read other published papers, where all details are present carefully, including reagent kit and instrument brand and manufacturer for qPCR. L480-485 and in other parts: Why ID of some germplasm accessions are underlined but other not? L493: Manufacturer and brand for Confocal microscope, please. L496: Ecotype of the used Arabidopsis thaliana, please. L499-501. “Using these two methods, Agrobacterium-mediated method and inflorescence infection method…” – this is unclear, which two methods authors indicated? In current form, this is only one method, for example, ‘floral deeping’ (or ‘painting of new open flowers’) of Arabidopsis plants with suspension of Agrobacterium containing the transgene. Please clarify. L506 (and L212-213) Please insert here and clarify what does mean “unload line (UL)’ used as controls? Is this Arabidopsis plants with ‘empty vector’ after genetic transformation or ‘escape’ (plants which did not contain the transgene after Agrobacterium transformation) or ‘null-segregants’ (plants with confirmed absence of the transgene, which occurred from progenies of hemizygote or heterozygote transgenic parents in T1 or T2? L507: Plants were “moved to nutrition bowl…”. What is this? On Figure 7A, plants are present growing in plastic square pots with substrate. ‘Nutrition bowl’ means a container under the pots for watering with nutrient solution via holes in the bottom of pots? What kind of substrate was used? Why these and other details are omitted? L521-523 and in other parts: Full names for all indicated genes are required in the first instance.

(3) Figure 9 and in other Figures. Please explain in M&M and in the legend of Figures, asterisks above columns were based on comparison of which WT, in controls or in treatment? The differences between transgenic plants and WT (for example in treatments) are very clear. However, what is about ‘UL’, which also used as controls together with WT? The differences between transgenic lines and ‘UL’ were insignificant in Figures 9B, 9E and 9F. If authors compared results of transgenic lines with only WT, which role was for ‘UL’? How authors can explain big differences in the treatment between WT and UL in the same Figures 9B, 9E and 9F? Please insert your explanation in the Discussion section.

Minor notes/corrections:

(4) In entire manuscript, starting from L16-17, authors used the term ‘resistance’ to abiotic stresses like salinity, cold, heat and other. However, this term is more suitable for biotic stresses like ‘disease resistance’ while ‘tolerance’ is much better for salinity and cold as authors correctly used in L33, L98 and in some other cases. I strongly recommend authors to use only ‘tolerance’ and ‘tolerant’ in this manuscript. Please keep in mind that ‘resistance’ and ‘tolerance’ have different biological meaning while in English linguistics they are often used as synonyms.

(5) L56 and L68. Full name ‘Transcription factors’ has to be use with the abbreviation (TF) in the first (but not in the second) occurrence. This is also true for ‘Jasmonic acid (JA)’ in L111 and L114.

(6) L81. There is no term “napus” but only ‘Brassica napus’ or ‘B. napus’.

(7) L552-554. Please either to include your Acknowledgements or delete this point with the template.

(8) Supplementary Table S1. Please insert a column with amplicon size for qPCR products and other products, where this is possible.

Round 2

Reviewer 2 Report

Authors made their great job and addressed most of the earlier issues but not all. Therefore, minor corrections listed below are still required to complete authors’ duty but no need attention from the reviewer.

Minor notes and corrections which are still must be addressed:

(1) L20, L542 and maybe in other parts. Please be consistent and use ‘Arabidopsis’ in Italics as a botanical name in Latin language and as it was correctly used in L21, L23, L26 and in many other spots.

(2) L80 and L140-141. Authors promised that they will correct ‘cold and salt tolerance’ but still are using ‘cold resistance’ and ‘cold and salt resistance’. Please be consistent and follow your own promise, correct and use only ‘tolerance’ rather than ‘resistance to cold, salinity and other abiotic stresses.

(3) L80. Botanical name ‘B. napus’ must be also in Italics.

(4) L287. Please correct units of salinity concentration. It has to be ‘200 mM NaCl’ with meaning ‘millimoles’ of concentration (as correctly used in L484) but not ‘200 mm NaCl’ with meaning ‘millimetres’ of a distance.

(5) L406-407. Please do not repeat two times ‘foreign gene’. Please replace one of the term for ‘transgene’ with the same meaning.

(6) L485. The phrase ‘dry drought’ is absurd. Please replace it for correct term ‘dehydration’. The following meaning ‘to simulate the drought conditions’ is correct.

(7) L540-541. Please correct the phrase “…was mediated by Agrobacterium-mediated method…” avoiding two times the identical term ‘mediated’. Additionally, ‘Agrobacterium’ must be in Italics as in Latin language.

(8) L605-607. Authors promised but did not change anything in the Acknowledgements. This only indicates that Authors did not read their revised manuscript properly.
